# Insights into the Genetic Architecture and Genomic Prediction of Powdery Mildew Resistance in Flax (*Linum usitatissimum* L.)

**DOI:** 10.3390/ijms23094960

**Published:** 2022-04-29

**Authors:** Frank M. You, Khalid Y. Rashid, Chunfang Zheng, Nadeem Khan, Pingchuan Li, Jin Xiao, Liqiang He, Zhen Yao, Sylvie Cloutier

**Affiliations:** 1Ottawa Research and Development Centre, Agriculture and Agri-Food Canada, Ottawa, ON K1A 0C6, Canada; chunfang.zheng@agr.gc.ca (C.Z.); nadeem.khan@agr.gc.ca (N.K.); pingchuan.li@agr.gc.ca (P.L.); liqiang.he@hainanu.edu.cn (L.H.); 2Morden Research and Development Centre, Agriculture and Agri-Food Canada, Morden, MB R6M 1Y5, Canada; khalid.rashid@agr.gc.ca (K.Y.R.); zhen.yao@agr.gc.ca (Z.Y.); 3Department of Biology, University of Ottawa, 30 Marie Curie, Ottawa, ON K1N 6N5, Canada; 4Key Laboratory of Crop Genetics and Germplasm Enhancement, College of Agriculture, Nanjing Agricultural University/JCIC-MCP, Nanjing 210095, China; xiaojin@njau.edu.cn

**Keywords:** powdery mildew (PM) resistance, quantitative trait loci (QTL), quantitative trait nucleotides (QTNs), genome-wide associate study (GWAS), genomic prediction (GP), core collection, flax, *Linum usitatissimum* L.

## Abstract

Powdery mildew (PM), caused by the fungus *Oidium lini* in flax, can cause defoliation and reduce seed yield and quality. To date, one major dominant gene (*Pm1*) and three quantitative trait loci (QTL) on chromosomes 1, 7 and 9 have been reported for PM resistance. To fully dissect the genetic architecture of PM resistance and identify QTL, a diverse flax core collection of 372 accessions augmented with an additional 75 breeding lines were sequenced, and PM resistance was evaluated in the field for eight years (2010–2017) in Morden, Manitoba, Canada. Genome-wide association studies (GWAS) were performed using two single-locus and seven multi-locus statistical models with 247,160 single nucleotide polymorphisms (SNPs) and the phenotypes of the 447 individuals for each year separately as well as the means over years. A total of 349 quantitative trait nucleotides (QTNs) were identified, of which 44 large-effect QTNs (*R*^2^ = 10–30%) were highly stable over years. The total number of favourable alleles per accession was significantly correlated with PM resistance (*r* = 0.74), and genomic selection (GS) models using all identified QTNs generated significantly higher predictive ability (*r* = 0.93) than those constructed using the 247,160 genome-wide random SNP (*r* = 0.69), validating the overall reliability of the QTNs and showing the additivity of PM resistance in flax. The QTNs were clustered on the distal ends of all 15 chromosomes, especially on chromosome 5 (0.4–5.6 Mb and 9.4–16.9 Mb) and 13 (4.7–5.2 Mb). To identify candidate genes, a dataset of 3230 SNPs located in resistance gene analogues (RGAs) was used as input for GWAS, from which an additional 39 RGA-specific QTNs were identified. Overall, 269 QTN loci harboured 445 RGAs within the 200 Kb regions spanning the QTNs, including 45 QTNs located within the RGAs. These RGAs supported by significant QTN/SNP allele effects were mostly nucleotide binding site and leucine-rich repeat receptors (NLRs) belonging to either coiled-coil (CC) NLR (CNL) or toll interleukin-1 (TIR) NLR (TNL), receptor-like kinase (RLK), receptor-like protein kinase (RLP), transmembrane-coiled-coil (TM-CC), WRKY, and mildew locus O (MLO) genes. These results constitute an important genomic tool for resistance breeding and gene cloning for PM in flax.

## 1. Introduction

Diseases are major constraints to crop production. Powdery mildew (PM) is a common fungal disease of many crops worldwide. Most PM pathogen species are host-specific or infect few hosts, implying that their genomes encode distinct “toolboxes” of pathogenesis-related genes [1]. Importantly, all PM fungi are obligate biotrophic plant pathogens; hence, their growth and reproduction are entirely reliant on the availability of water and nutrients from living host cells [2]. PM interferes with plant growth and reduces the crop’s quality. For example, PM can reduce grain yields by up to 40% in wheat [3]. PM occurs later in the season when temperatures are between 20 and 25 °C, and the extent of the damages is only determined after the crop has been harvested [4]. Once the host is infected, the infection spreads quickly under favourable conditions, resulting in PM outbreaks. Furthermore, the evolution of pathogen races with increased virulence may result in a “breakdown” of resistance. For instance, *Pm17*, *Pm3a*, and *Pm4a* were defeated in several Eastern and mid-Atlantic regions of the United States [5,6], as was *Pm8* in China [7]. As a result, most resistance genes become ineffective after a period of time [8]. Thus, it is necessary to keep looking for new sources of resistance or create new combinations to stay abreast of new races. This can be achieved by identifying new genes/alleles that will allow us to build molecular tools to quickly and efficiently introduce them into breeding lines and to diversify the resistance sources against the rapidly evolving pathogen races.

Plant disease resistance is usually categorized as either “qualitative,” defined by the presence of major resistant (*R*) genes, or “quantitative” defined by the presence of resistance-related quantitative trait loci (QTL) [9,10]. *R* genes, for example, have been used successfully in wheat where they form the basis of resistance breeding programs that have produced many resistant commercial varieties [11]. The majority of the known R proteins have nucleotide-binding site (NBS) and leucine-rich repeat (LRR) domains and act as intracellular immune receptors that recognize their cognate effectors directly or indirectly [12]. Two other genetically specified R proteins are cell surface-localized receptor-like transmembrane proteins (RLPs) and receptor-like kinases (RLKs) [13]. Mildew locus O (MLO) is a well-studied PM susceptibility gene that was first discovered in barley in 1942 [14] and later identified in rice and wheat [15,16].

Qualitative resistance means “monogenic”, “vertical” or “race-specific”, while quantitative resistance refers to adult plant resistance (APR), “slow-mildewing”, or “partial resistance” [17]. Qualitative resistance genes are usually short-lived due to the frequent changes in the pathogen population [9]. Quantitative genetics approaches, such as estimating genetic elements, heritability, and efficient gene numbers, are typically used to investigate quantitative resistance. Several studies have identified more than 100 QTL on all chromosomes in wheat [11]. Quantitative resistance is more durable [18,19], and offers long-term defence to host plants. However, detecting such resistance is difficult, particularly when *R* genes are present. Traditionally, genetic studies of quantitative traits use segregating biparental populations that have been tested for the traits of interest and genotyped with DNA-based molecular markers. This method necessitates the costly and time-consuming creation of specific mapping populations. Additionally, the resolution is limited by the number of crossovers and the high linkage disequilibrium (LD), necessitating additional research to fine-map the QTL that often span several cM [20].

Genome-wide association studies (GWAS) are a powerful alternative strategy to the traditional linkage-based QTL mapping, focusing on LD obtained from unrelated genotypes of a collection that reflects historical recombination, thereby resulting in more accurate positioning of QTL and higher mapping resolution. However, GWAS are less effective at detecting alleles with small effects and rare alleles, and can produce a large number of false-positive associations. To improve the statistical mapping power of GWAS, a large population size and high marker density are needed. Several GWAS analyses for PM resistance have been performed in plant species such as wheat [11,21,22], barley [23] and oat [24]. These studies revealed that resistance to PM was mainly mediated by a few minor QTL with small effects and heavily influenced by the genetic background of the populations studied, the phenotyping conditions, and the genotype-by-environment interactions. These findings imply the limited efficiency of marker-assisted selection (MAS) to improve PM resistance by pyramiding small-effect favourable alleles. Genomic selection (GS), also known as genomic prediction (GP), is a promising alternative in crops, particularly for improving complex traits. In GS, genome-wide markers are used to predict the genomic-estimated breeding values (GEBVs) of individuals by capturing the benefits of both major- and minor-effect QTL [25,26]. As a result, GS captures a more significant proportion of the genetic variance of the selected traits than MAS, which is often restricted to a small number of markers linked to major QTL.

In flax, PM, caused by the obligate biotrophic ascomycete *Oidium lini* Skoric, is one of the most destructive and common flax foliar diseases [27,28]. It was first reported in Western Canada in 1997 [29] where it remains a sporadic disease. Infection with PM at an early stage of development will result in significant yield and seed quality reductions [30]. Most Canadian flax varieties are moderately resistant to PM under field conditions with natural inoculum [31]. One major dominant gene for resistance to PM, designated *Pm1*, has been identified from the Canadian varieties AC Watson, AC McDuff and AC Emerson as well as from the introduced varieties Atalante and Linda. Two additional dominant genes have also been postulated in Linda [32]. The use of resistant varieties in combination with a systematic disease management program is the most successful way to reduce the incidence of PM and save producers money.

In the present study, both GWAS and GS analyses were performed on 447 flax accessions. Field resistance assessment was performed for five to eight years (2010–2017) in Morden, Manitoba, Canada and genotyping was achieved through short-read sequencing of genomic DNA. The major objectives of this study were: (1) to discover major and minor QTNs associated with PM resistance in flax through GWAS using seven widely used multi-locus and two single-locus statistical methods; (2) to identify putative candidate genes for these QTNs; and (3) to evaluate the efficiency of the QTNs as markers in GS models.

## 2. Results

### 2.1. Evaluation of Powdery Mildew Resistance

The average PM rating of the 372 accessions of the core collection over five years (2012–2016) was 4.84 ± 1.35, while that of the 75 selected breeding lines was 2.24 ± 0.43 (Figure 1A, Appendix A). The overall average was 4.5 ± 1.4 over the five years with large phenotypic variation (coefficient of variation (CV): 28.1–53.2%) (Table 1).

Based on the average PM ratings over five years, 85 genotypes were either highly resistant (HR) or resistant (R); this group included 70 of the 75 selected breeding lines and 15 accessions from the core collection. Of these 85 genotypes, 83 belong to the linseed morphotype while F_LTU_B_CN10111 and F_NLD_C_CN18983 are fibre-type flax. A total of 92 genotypes were moderately resistant (MR), including the five selected breeding lines; 114 genotypes were moderately susceptible (MS); 118 genotypes were susceptible (S); and 38 genotypes were highly susceptible (HS) (Figure 1B). The PM ratings averaged 2.2 ± 0.4, 3.4 ± 0.3, 4.4 ± 0.3, 5.8 ± 0.6 and 7.3 ± 0.3 for the R, MR, MS, S and HS groups, respectively (Figure 1B).

Similar performance for PM resistance between years was observed, although the yea-to-year PM ratings of the genotypes differed slightly. The PM ratings in 2014 and 2015 were generally lower than those in other years (Table 1), while the PM ratings of the 75 breeding lines were lower in 2010 (Appendix A). Pearson correlation analysis corroborated the significant correlations across years (*r* = 0.35–0.81, *p* < 0.01) (Figure 2).

### 2.2. Genetic Structure of the Population

Principal component analysis (PCA) was performed for the flax core collection of 372 accessions and 75 selected breeding lines using all 247,160 SNPs identified from the 447 genotypes (Appendix A). The first three principal components (PCs) accounted for 11.8%, 8.1% and 6.1% of the total variation, respectively. These three PCs grouped all genotypes into either linseed or fibre morphotypes (Appendix A). Most of the 75 selected breeding lines were highly resistant to PM (Figure 1A), creating a spatial cluster that differed from the core collection (Appendix A). To capture more of the structural contributions of the PCs, the first ten PCs (38% of the total variation) were selected as a genetic structure matrix for the downstream GWAS analyses.

### 2.3. Identification of QTL

Two single-locus methods (GLM and MLM) and seven multi-locus methods (pLARmEB, pKWmEB, FASTmrMLM, ISIS EM-BLASSO, mrMLM, FASTmrEMMA, and FarmCPU) were used to identify PM resistance QTNs from all 247,160 SNPs and all 447 accessions. A total of 349 unique QTNs were identified for the six PM datasets (PM-2012, PM-2013, PM-2014, PM-2015, PM-2016, PM-Mean) based on the 47,564 haplotype blocks identified from the 247,160 SNPs using Plink (Appendix A). QTNs located in the same haplotype blocks were treated as QTL, and the QTNs with the largest effect (*R*^2^) were chosen as the tag to represent these QTL. Singleton QTNs were considered independent QTL. Hereafter, we use the tag QTNs or singleton QTNs to represent the QTL. The 349 tag QTNs and their related information are listed in Appendix A and depicted on chromosomes in Figure 3.

The 349 unique QTNs were identified using a number of GWAS statistical models (Appendix A). The single-locus model GLM detected only nine QTNs, but these had relatively large effects, ranging from 0.65 to 27.62% and averaging 11.34% of *R*^2^ (Appendix A). No QTNs were detected with the single-locus model MLM. The multi-locus FarmCPU method detected 15 QTNs with relatively large effects (9.07% of *R*^2^). The remaining six mrMLM-based multi-locus models identified most of the QTNs, including some with large and minor effects accounting for 3.19–5.79% of *R*^2^. The QTN numbers varied from 68 with FASTmrEMMA to 108 with pLARmEB (Appendix A). Only 2–4 QTNs were common between GLM and the seven multi-locus models; in contrast, 13–39 QTNs were shared by two of seven multi-locus models (Appendix A).

The identified QTNs also differed across individual PM phenotypic datasets (Appendix A). A total of 45–84 QTNs were identified from the six PM datasets, of which few were shared by more than one PM dataset (Appendix A). However, when assessing the effects of the QTN alleles across PM datasets, most QTNs were found to be stable over years. Of the 349 unique QTNs, 265 had significant allele effects in at least three of the PM datasets (Appendix A). A total of 122, 56, 54, 33, 32 and 52 QTNs had significant allele effects in six, five, four, three, two and one PM datasets, respectively. QTN effect *R*^2^ estimates were positively correlated with the number of PM datasets with significant allele effects and negatively correlated to the coefficients of variation (CVs) of the QTN effects, indicative of the stability of the QTNs (Figure 4). Large-effect QTNs were stable across years, while the small-effect QTNs only significant in one PM dataset were least stable (Figure 4, Appendix A). Of the 122 QTNs stable across all six PM datasets, 44 were of large QTN effects ≥ 10% of *R*^2^, of which the following 11 had *R^2^* values ≥ 20%: *Lu2-1672205* (30.2%), *Lu3-581507* (29.71%), *Lu5-11130392* (27.8%), *Lu5-12090990* (27.6%), *Lu5-15697144* (26.9%), *Lu5-16602027* (25.2%), *Lu5-16840013* (22.6%), *Lu7-17007593* (21.3%), *Lu9-3920670* (20.9%), *Lu9-20701159* (20.9%), *Lu11-17188390* (20.7%).

Most large-effect QTNs were clustered on the distal ends of chromosomes, especially on chromosome 5 (0.4–5.6 Mb and 9.4–16.9 Mb) and 13 (2.6–4.9 Mb) (Figure 3, Appendix A). Of the 44 large-effect QTNs with *R*^2^ ≥ 10%, 15 were located on chromosome 5 and 5 on chromosome 13.

### 2.4. Favourable Alleles

According to the effect direction (positive or negative) of two alleles for a QTN, the favourable allele (FA) composition of all identified QTNs existing within each genotype was determined. For all 349 unique QTNs, the total number of favourable alleles (NFAs) within a genotype ranged from 141 to 305. The NFAs were significantly negatively correlated with the PM ratings of the accessions (Figure 5A) (*R*^2^ = 0.62). The 75 selected breeding lines that were resistant (R) or moderately resistant (MR) were clearly distinguished through NFAs of accessions. The NFAs for five groups of genotypes were 262 ± 35, 204 ± 22, 193 ± 15, 180 ± 13 and 169 ± 12 for R, MR, MS, S and HS, respectively (Figure 5B), confirming the same linear correlation between NFAs and PM ratings displayed in Figure 5A.

Based on the number of favourable alleles in the genotypes of the population, three types of QTNs were observed (Figure 6, Appendix A): type 1 QTNs with low favourable allele frequencies (FAFs) (Figure 6A), type 2 QTNs with high FAFs (Figure 6B), and type 3 QTNs with FAFs close to 0.5 (Figure 6C). Of the 447 genotypes in this study, 177 or 37% were R or MR; thus, the FAFs of good QTNs were expected to be approximately 0.37. We found that the 33 QTNs with *R*^2^ > 20% had an average FAF of 0.27 (0.15–0.33), and the 33 QTNs with *R*^2^ between 10% and 20% had an average FAF of 0.48 (0.19–0.86), compared to the remaining QTNs with *R*^2^ ≤ 10% that had an average FAF of 0.60 (0.06–0.94). The FAFs of most large-effect QTNs (>10%) ranged from 0.1 to 0.6 in all genotypes and 0.5 to 0.9 in resistant genotypes (Figure 7).

### 2.5. Relationship between PM Resistance and Flax Morphotypes

Of the 447 accessions used in this study, 367 and 80 belonged to the linseed and fibre morphotypes, respectively. Linseed and fibre genotypes had average PM ratings of 4.2 ± 1.5 and 5.3 ± 1.5, and average NFAs of 208.8 ± 38.1 and 175.7 ± 12.2, respectively (Figure 8). Significant differences in PM ratings and NFAs between linseed and fibre genotypes were observed (Wilcox test, *p* = 2.487 × 10^−8^ for PM rating and *p* < 2.2 × 10^−16^ for NFAs). The 10% most resistant accessions were all linseed types, while the 10% most susceptible included 18 fibre accessions (~38%) even though fibre accessions made up only 17.9% of the overall collection. Overall, a higher proportion of fibre accessions than expected based on their representation in the collection were susceptible, and this was congruent with the proportionally lower NFAs.

### 2.6. Candidate Genes

Of the 349 unique QTNs, 93 were located within genes (Appendix A), of which 6 were RGAs: *Lus10002249* on Lu8, *Lus10015649* on Lu14, *Lus10003971* on Lu7, *Lus10006772* (RLK) on Lu12, and *Lus10030587* and *Lus10033608* (RLK) on Lu12. *Lus10002249* and *Lus10015649* are Toll/interleukin-1 receptor (TIR)-nucleotide-binding site (NBS)-leucine-rich repeats (LRR) (TNL) genes, and *Lus10003971* is a transmembrane (TM) coiled-coil (CC) (TM-CC) gene, while the remaining three encode receptor-like protein kinase (RLK).

To further identify candidate genes for PM resistance, a subset of 3230 SNPs located on 838 flax RGAs were extracted from the overall SNP dataset and GWAS analysis was performed with the same models as mentioned above. Significant QTNs were identified from 42 RGAs, including *Lus10030587*, *Lus10003971*, and *Lus10002249* that had already been detected with the 247,160 genome-wide SNP dataset. Combining the results from both datasets, a total of 388 QTNs were obtained, including 132 QTNs identified on 45 RGAs (Table 2) and 87 non-RGA genes (Appendix A).

Of these 388 QTNs, 269 harboured 445 RGAs within 200 Kb regions of the QTNs (Appendix A), encompassing 13 gene families: dirigent protein (DIR), disease resistance-zinc finger-chromosome condensation (DZC), extreme-drug-resistant (EDR), mildew resistance locus o (MLO), RLK, receptor-like protein (RLP), resistance to powdery mildew 8 (RPW8), TM-CC, TIR, nucleotide-binding site–leucine-rich repeats (NL), coiled-coil–nucleotide-binding site–leucine-rich repeats (CNL), TNL and WRKY (Appendix A). Of these 445 candidate RGAs, 45 had QTNs identified within them, and 270 RGAs were supported by at least one SNP on each gene, which had significant SNP allele effects (Appendix A). Candidate genes for large-effect QTNs were mostly TNL, CNL, WRKY, RLP and RLK genes (Table 3). For the 269 QTN regions harbouring RGAs, the average minimum distance between a QTN and a predicted candidate RGA was 32,346 bp with a range of 0–99,046 bp.

A total of 16 candidate RGA clusters that contained at least 3 candidate genes were observed on 11 of the 15 flax chromosomes (2, 3, 5, 7–10, 12–15) (Appendix A). The largest candidate gene cluster was associated with QTN *Lu8-18351964* (*R*^2^ = 11.49%) located on Chr 8. It contained 15 tandemly duplicated TNL genes within a 126.5 Kb region (18,254,394–18,380,935 bp). The QTN *Lu8-18351964* was identified within gene *Lus10007812*, while the remaining 14 genes had at least one SNP per gene with a significant allele effect on PM resistance. Another important gene cluster was associated with QTNs *Lu5-1534998* (*R*^2^ = 27.83%) and *Lu5-1535619* (*R*^2^ = 32.06%) and spanned a genomic region of 123.5 Kb (1,449,598–1,573,096 bp) containing four TNL genes: *Lus10004726*, *Lus10004727*, *Lus10004719* and *Lus10004747*. The QTN *Lu5-1535619* was identified within the coding region of *Lus10004726*, but the other three genes also had SNPs with significant allele effects on PM resistance (Appendix A).

With the exception of *Lus10040576*, *Lus10021001*, and *Lus10031043*, the remaining 42/45 candidate RGA genes that co-located with QTNs were considered highly stable over years with CV < 100% (Table 2). These 45 RGAs belonged to the following gene families: TNL (16), RLK (16), WRKY (4), TM-CC (4), RLP (2), CNL (1), DIR (1), and MLO (1). Among those with the highest *R*^2^ (>20%) (Appendix A), *Lus10004727*, *Lus10004726* and *Lus10004719* represent tandemly duplicated TNL genes (*R*^2^ = 24–28%); *Lus10029860* is another TNL gene (*R*^2^ = 35%); and *Lus10032303* encodes a WRKY gene (*R*^2^ = 24%). All five genes were located on chromosome 5 (1.5–13.3 Mb). The large-effect candidate RGA *Lus10027903* on chromosome 12 encodes an RLP gene (*R*^2^ = 20%).

A notable exception is the 108.6 Kb genomic region of chromosome 13 (Figure 9), which harboured three tandem duplicate *RPW8* orthologous genes (*Lus10000835*, *Lus10000836* and *Lus10009328*). Indeed, we identified four major QTNs at this locus, *Lu13-4791823* (*R*^2^ = 13.79%), *Lu13-4830850* (*R*^2^ = 7.17%), *Lu13-4866704* (*R*^2^ = 12.24%), and *Lu13-4900476* (*R*^2^ = 10.9%), but no SNPs were found within the three *RPW8* orthologous genes, probably due to genome sequencing resulting in the low read coverage in the region. An alternative reason could be that the reads did not precisely map to the three orthologues and that the SNPs were not called because they did not pass the filtering criteria.

From the candidate RGA genes, 77 gene pairs were duplicated between or within chromosomes due to whole-genome duplication (Appendix A). Duplicated genes involved 131 QTNs on all 15 chromosomes (Figure 3, Appendix A). Out of 44 large-effect QTNs (*R*^2^ > 10%), 10 (*Lu3-1644588*, *Lu4-11526385*, *Lu5-1552921*, *Lu5-1763832*, *Lu5-15697144*, *Lu7-17007593*, *Lu9-4948236*, *Lu10-11682031*, *Lu10-11695343*, *Lu15-46304*) involved 13 pairs of duplicated genes, including 1 pair each of TNL genes, CNL genes, MLO genes, TM-CC genes and WRKY genes as well as 8 pairs of RLK genes (Figure 3, Appendix A). Some QTNs harboured two or more candidate genes, and thus involved more than one pair of duplicated genes.

### 2.7. Genomic Prediction for PM Resistance

To evaluate the overall reliability of the identified QTNs, ten GS models were compared using the 349 unique QTNs and a five-fold cross-validation scheme was used to identify the best models for genomic prediction of PM resistance. All models performed similarly, except for RKHS and RFR (Appendix A). GBLUP, BRR and SVR generated the highest predictive ability of 0.93. Thus, GBLUP was used for the remaining comparisons.

Five marker sets were used to construct the GBLUP GS models: (1) 349 QTNs identified from the genome-wide dataset of 247,160 SNPs, (2) 388 QTNs obtained using the genome-wide and the RGA-derived SNP datasets, (3) 132 QTNs located within genes, including 45 RGAs and 87 non-RGA genes, (4) 294 QTNs or SNPs located on 294 candidate RGAs with significant allele effects, and (5) the genome-wide dataset of 247,160 SNPs. The 349 QTN and the 388 QTN datasets explained the highest genetic variation for PM resistance (*h*^2^ = 0.69–0.70) and generated the highest predictive ability (*r* = 0.925–0.926) (Table 4). The predictive ability obtained from the other datasets was significantly lower and the lowest was obtained from the genome-wide dataset of 247,160 SNPs (*r* = 0.690).

The prediction model constructed using all 447 accessions as a training population, the PM ratings over five years, and all 349 QTNs with GBLUP explained 96% of PM variation (*R*^2^ = 0.96), showing high predictive ability and the potential of this model in applied genomic prediction (Appendix A).

## 3. Discussion

Genetic studies of PM resistance in flax are few in number, limiting our understanding of the genetic architecture of this trait to a few major genes. Indeed, using traditional genetic analyses, the single dominant gene *Pm1* that confers resistance to PM was identified in several Canadian (‘AC Watson’, ‘AC McDuff’, and ‘AC Emerson’) and introduced (‘Atalante’ and ‘Linda’) cultivars, and two putative dominant genes were additionally postulated in ‘Linda’ [32]. In agreement with the latter, a QTL mapping study using biparental F_3_ and F_4_ populations derived from a cross between the susceptible cultivar NorMan and the resistant cultivar Linda identified three PM resistance QTL [27]. Here, we exploited a large genetic panel that included 372 accessions from the diverse flax core collection [33,34] from which we defined a high density genome-wide dataset of 247,160 SNPs and an RGA-specific subset of 3230 SNPs, with the view of gaining greater insights into the genetic architecture of PM resistance in flax in order to design breeding methods for its improvement. Because the core collection contained few highly resistant lines, we augmented the germplasm with 75 selected breeding lines previously phenotyped and deemed highly resistant to PM. This significantly enhanced the detection power and the reliability of the identified QTNs through an increase in population size and the genetic variation of the PM ratings. We also used two single- and seven multi-locus statistical models to identify both large- and small-effect QTNs, resulting in a total of 349 unique QTNs from the genome-wide SNP dataset (Appendix A). This has proven to be a good strategy to take advantage of the strength of each model as well as to palliate their shortcomings [35,36]. Post-identification QTN analysis (Appendix A) also contributed to the reliability of the QTNs by removing potentially redundant and false-positive QTNs. Overall, the methodology used herein constitutes a powerful strategy, combining different tools and methods to identify the most reliable QTN–trait association.

Three PM-resistant QTL located on LG1 (*QPM-crc-LG1*), 7 (*QPM-crc-LG7*), and 9 (*QPM-crc-LG9*) were identified using biparental F_3_ and F_4_ families derived from a cross between NorMan (PM-susceptible) and Linda (PM-resistant) [27]. The SSR markers defining these three QTL correspond to three genomic regions of 16,920,407–18,739,647 bp on Chr 1, 3,817,603–3,817,863 bp on Chr 7, and 357,191–357,510 bp on Chr 9 [37]. The genomic region of *QPM-crc-LG1* spanned two QTNs identified in this study, *Lu1-18139539* and *Lu1-18452203*, of which *Lu1-18139539* was detected within a gene *Lus10009706* (coordinated from 18,138,555 to 18,140,489 bp on Chr 1), encoding a tetratricopeptide repeat (TPR)-like superfamily protein. However, no QTNs identified here overlapped with *QPM-crc-LG7* and *QPM-crc-LG9*. Because the *Pm1* gene [32] was identified using phenotypic data only, we were not able to map this gene to a chromosome.

Current GWAS methods are limited to predicting genes or genetic features controlling traits. This may mostly depend on the density of genome-wide markers, the size of a genetic panel, the association of QTN with the trait or the extent of the QTN effect, and so on. To date, the main method for predicting candidate genes from QTNs identified by GWAS remains to scan the annotated genes in their vicinity. An optimal window size may be determined based on linkage disequilibrium decay [38,39]. In this study, we adopted a window of 200 Kb flanking a QTN [35,40]. To provide further support to the candidate RGAs as it relates to their function in disease resistance, we first narrowed the candidate genes to the flax RGAs identified from its annotated reference genome sequence [37,41,42]. Second, we reduced the set to the candidate RGAs located within the specified window size of 200 Kb and that had at least one SNP with significant allele effects on PM ratings within the RGAs. That is, even though no QTN was identified from a candidate gene, the SNP(s) on a candidate gene must have been significantly related to PM rating to be considered. Third, we performed GWAS using the 3230 SNPs exclusively located within flax RGAs in order to identify QTNs associated with PM resistance that were specifically located within RGAs. In this way, we identified 45 RGAs that possessed intragenic QTNs and an additional 270 RGAs supported by SNPs within these genes, all with significant allele effects.

QTL mapping provides a useful statistical genetics tool to identify QTNs and candidate genes associated with PM resistance. Molecular markers can be designed from some large-effect QTNs for MAS. Alternatively, or in addition to, all QTN markers can be used to establish GP models for predicting GEBVs of germplasm or breeding lines. In this study, we counted favourable alleles of all QTNs (ignoring QTN effects) in each accession of the genetic panel and we observed a significant correlation between the number of favourable alleles and PM ratings (*R*^2^ = 0.62, Figure 5A). PM-resistant accessions had significantly more favourable alleles than PM-susceptible accessions (Figure 5B), clearly indicating the significant additive feature of the identified QTNs. Consequently, the pyramiding of favourable alleles through cross prediction [43], hybridization and recombination, as well as GS is expected to translate into cultivars with improved resistance to PM. The 75 selected breeding lines with high PM resistance and a high number of favourable alleles demonstrate the potential of this approach. Genomic cross prediction is an advanced genomic tool that integrates computer simulation and GS to predict the genetic performance of different types of crosses by evaluating the expected breeding values and genetic variances of their segregating populations for the purpose of selecting superior crosses and consequently enhancing the potential for success [43]. GS has been evaluated via a cross-validation approach for agronomic, abiotic and biotic stress-related traits, including pasmo resistance in flax [36,44,45]. For example, the predictive ability of pasmo resistance in flax was 0.92 when 500 QTL were used for prediction [44], similar to our results of 0.925 obtained using 349 QTNs for PM resistance. Therefore, these QTNs offer the potential for PM resistance breeding using genomics-assisted breeding methods.

The potential candidate genes can be further validated using functional approaches. Once functionally validated, they can be genetically edited to improve cultivar PM resistance. For example, *MLO* is a resistance gene that confers PM resistance in many crops, such as grapevine [46], wheat [47], and barley [48]. Several technologies such as genome editing and targeting induced lesions in genomes (TILLING) have been used to create MLO mutants with enhanced PM resistance in bread wheat [47,49]. Nekrasov et al. (2017) [50] reported a non-transgenic tomato variety resistant to PM (*Oidium neolycopersici*) produced through editing the *MLO* gene (*SlMlo1*) using the CRISPR/Cas9 technology, which is based on the Cas9 DNA nuclease guided to a specific DNA target by a single guide-RNA (sgRNA). *PMR4* encodes a callose synthase, and its loss-of-function mutants are resistant to PM in *Arabidopsis* and tomato. The CRISPR/Cas9-mediated knockout mutants of the *PMR4* ortholog (*SlPMR4*) in tomato showed partial resistance against the PM pathogen *O. neolycopersici* [51]. RNA silencing of *SlPMR4* also enhanced the resistance to PM in tomatoes [52]. In this study, we detected six MLO gene orthologues (*Lus10036120*, *Lus10036121*, *Lus10023506*, *Lus10015461*, *Lus10012698*, and *Lus10001336*) that had significant SNP effects with *R*^2^ of 1–10%. QTN *Lu13-1749576* (*R*^2^ = 1.23%) was identified from the gene region of the *MLO* ortholog *Lus10001336*. These MLO and other identified candidate genes could be candidates for genome editing to improve PM resistance.

The *RPW8* domain was found in several broad-spectrum powdery mildew resistance proteins from *Arabidopsis thaliana* and other dicots [53,54]. The *A. thaliana* locus *RPW8* contains two naturally polymorphic, dominant *R* genes, *RPW8.1* and *RPW8.2*, which individually control resistance to a broad range of powdery mildew pathogens [53]. Therefore, the three *RPW8* genes identified on chromosome 13 are potentially important candidate genes. An additional PM resistance study based on a fibre flax recombinant inbred line (RIL) population derived from the Aurore/Adelie cross also showed that several *RPW8* genes may explain the PM resistance difference between the two parents [55].

It is worth noting that some paired candidate duplicate genes co-located with some paired QTNs (Appendix A). For instance, in Figure 9, three *RPW8* genes (*Lus10000835*, *Lus10000836* and *Lus10009328*) are paired duplicate genes with sequence similarity of 0.67–0.69 (Appendix A). Correspondingly, four QTNs *Lu13-4791823*, *Lu13-4830850*, *Lu13-4866704*, and *Lu13-4900476* are paired QTNs. This finding is worthy of further investigations to test whether these duplicated genes and QTNs contribute additively to PM resistance in flax.

We observed that in the combined population of 372 accessions from the core collection and 75 breeding lines, linseed accessions tended to be more resistant to PM than fibre accessions. The same result was also observed in the flax core collection alone [56]. This result is somewhat surprising because fibre flax is grown in coastal, humid environments that are conducive to the development of the disease and resistance to PM has long been a major objective of fibre flax breeding programs considering that PM has the potential to considerably reduce fibre quality. Therefore, crosses with resistant linseed varieties can be used to introduce new allelic diversity for PM resistance in fibre flax breeding programs and genomic selection can be used to assist in selecting the most resistant lines while preserving the fibre morphotype.

## 4. Materials and Methods

### 4.1. Genetic Population

A diverse genetic panel of 372 accessions from the previously described flax core collection [33,34] was used. The core collection was assembled from the world collection of 3378 flax accessions, collected from 39 countries and corresponding to 11 regions of the world: North America, South America, Eastern Asia, Western Asia, Southern Asia, Central and Eastern Europe, Western Europe, Southern Europe, Northern Europe, Oceania, and Africa. This panel contained 17 landraces, 84 breeding lines, 234 cultivars, and 37 accessions of unknown improvement status that were grouped into two morphotypes: 80 fibre and 292 linseed accessions [56].

The core collection [56] contained few highly resistant accessions. To empower the GWAS and GP analyses, an additional 75 resistant breeding lines were added to the core collection.

### 4.2. Phenotyping of Powdery Mildew Resistance and Statistical Analysis

The 372 accessions were evaluated for reaction to PM at Agriculture and Agri-Food Canada, Morden Research and Development Centre’s farm, Morden, Manitoba, Canada from 2012 to 2016. The experimental design was a randomized, complete block design with two replications. Each entry was seeded in 3 m rows spaced 30 cm apart during the 2nd or 3rd week of May every year. Inoculated susceptible plants were transplanted from the growth room into the field at the early flowering stage to serve as inoculum and ensure early disease infection and development in the field. One pot containing ten heavily infected plants was transplanted every ten rows. Disease ratings (PM severity) on leaves and stems were evaluated as the percentage of the leaf and stem areas covered by mycelium using the following 0 to 9 scale: 0 (HR) = no sign of PM—most vigorous plants, 1 (HR) = <10%, 2 (R) = 11–20%, 3 (MR) = 21–30%, 4 (MS) = 31–40%, 5 (S) = 41–50%, 6 (S) = 51–60%, 7 (S) = 61–70%, 8 (HS) = 71–80%, and 9 (HS) ≥ 80% [32]. This rating was converted into five resistance groups: (1) resistant (including highly resistant HR and resistant R, rating 1–2.9); (2) moderately resistant (MR, rating 3.0–3.9); (3) moderately susceptible (MS, rating 4.0–4.9); (4) susceptible (S, rating 5.0–6.9); and (5) highly susceptible (HS, rating 7.0–9.0). Field assessments were conducted at the early (PM1, 26 July) and late flowering stages (PM2, 7–10 days after PM1, around 2 August), the green boll stage (PM3, 7–10 days after PM2, around 10 August), and the early brown boll stage (PM4, 7–10 days after PM3, around 20 August). In 2012 and 2013, the observations from all four stages (PM1–PM4) were collected. Due to weather and other factors, only the data from PM1 and PM2 were available in subsequent years.

The 75 selected breeding lines were field-evaluated using the same procedure as the flax core collection, but this was accomplished during eight consecutive years (2010–2017). As both populations were assessed in the same PM nursery using the same procedure, the two datasets of the five common years (2012–2016) were combined for downstream analyses.

### 4.3. Genotyping and SNP Identification

Whole-genome resequencing methodology was employed to genotype all individuals of the core collection. The lines were grown in growth chambers with 20 h light at 22° and 4 h dark at 18° until they were approximately 7–8 cm tall. The compact tip of the plants (75–100 mg) was collected, flash-frozen in liquid nitrogen, and immediately lyophilized. Genomic DNA was extracted using the DNeasy 96 Plant kit (Qiagen, Mississauga, ON, Canada) and quantified using the Quant-iT PicoGreen dsDNA assay kit (Thermo Fisher Scientific, Waltham, MA, USA), both according to the manufacturer’s instructions. The Illumina HiSeq 2000 platform (Illumina Inc., San Diego, CA, USA) was used to generate 100 bp paired-end reads with ~15.5X genome coverage equivalents of the reference genome. All reads from each individual of the population were aligned to the scaffold sequences of the flax reference genome [57] using BWA v0.6.1 [58] with base-quality Q score in Phred scale >20 and other default parameters. The alignment file for each individual was used as input for SNP discovery using the software package SAMtools v1.12 [59]. All variants were further filtered to obtain a set of high-quality SNPs as previously described [60]. The SNP coordinates were then converted to the chromosome scale of the flax pseudomolecules v2.0 upon its release [42]. All procedures were implemented in the AGSNP pipeline [61,62] and its updated GBS version [60]. The detected SNPs were further filtered with minor allele frequency (MAF) > 0.05 and SNP call rate ≥ 60%. To minimize the contribution from regions of extensive strong linkage disequilibrium (LD), a single SNP was retained per 200 Kb window when pairwise correlation coefficients (*r^2^*) among neighbouring SNPs were greater than 0.8 [63,64], resulting in a total of 258,873 SNPs. Missing SNPs (on average, 14.13% of a missing data rate) were imputed using Beagle v.4.2 with default parameters [65].

A similar approach was used for the genotyping of the 75 selected breeding lines. Shotgun PCR-free library preparation (Lucigen, Middleton, WI, USA), library quality control (Illumina, San Diego, CA, USA) and sequencing were performed by the Centre d’expertise et de services Génome Québec (Montréal, QC, Canada). Twenty-one samples were indexed per lane, and sequencing on the HiSeqX platform (Illumina) generated 150 bp paired-end reads to an estimated average of 15X genome coverage per genotype. The same procedure was used to identify SNPs from the 75 selected breeding lines. The identified SNPs were combined with the SNP set from the core collection, resulting in a common set of 247,160 SNPs that was used in all downstream analyses.

### 4.4. Genome-Wide Association Studies (GWAS)

GWAS analyses were conducted separately for the five individual year datasets and the 5-year average dataset with the single-locus models GLM [66] and MLM [67] and the following seven multi-locus models: FarmCPU [68], pLARmEB [69], pKWmEB [70], FASTmrMLM [71], FASTmrEMMA [72], ISIS EM-BLASSO [73] and mrMLM [71]. The R package rMVP v1.0.4 [68] was used to run the GLM, MLM and FarmCPU models, while the R package mrMLM v5.0.1 [74] was used to run the remaining six multi-locus models. The kinship matrix used for each model was calculated using the module implemented in the corresponding software. The population structure of the 447 accessions was estimated using principal component analysis (PCA) and the first ten principal components (PCs) as cofactors in the models.

The threshold of significant associations for GLM, MLM and FarmCPU was determined by a critical *p* value (α = 0.05) subjected to Bonferroni correction, i.e., the corrected *p*-value = 2.02 × 10^−7^ (0.05/247,160 SNPs). A log of odds (LOD) score of 3.0 was used to detect significant associations for the six models implemented in the mrMLM package.

The putative QTNs identified were first analysed by testing the statistical significance of QTN alleles for PM resistance. Statistically significant differences between alleles provided validity to the QTNs. Wilcox non-parametric tests were performed using the R function *wilcox.test* to remove the non-significant QTNs at a 5% probability level. The direction (positive or negative) of QTN effects was subsequently determined. Only QTNs with consistent effect directions in all datasets were considered valid and were retained. Such QTNs were grouped into QTL by calculating haplotype blocks using plink v1.9 [75]. QTNs located in the same haplotype block were grouped into QTL. For each such defined QTL, the QTN with the largest average *R*^2^ over all datasets was chosen as the tag QTN representative of the QTL. *R*^2^ values were calculated based on simple regressions of QTNs on PM ratings, representing the proportion of the total variation for PM resistance explained by the QTNs/QTL.

To analyse the stability of the QTNs, we used the coefficient of variation (CV) of the allele effect values across the six PM datasets (five individual years and mean over years) for each QTN. Favourable alleles were determined based on the difference in PM rating of individuals with either of the two alleles at a given QTN. Stable QTNs have the same favourable allele in all PM datasets. QTNs with inconsistent effect values were considered unreliable and were removed. QTNs were declared highly stable if the effect differences across datasets were significant. Thus, for a significant QTN, its effect difference must be significant in at least one PM dataset. CV for effect values over datasets can be used to measure the stability of QTNs. The second criterion was *R*^2^ (%), i.e., the variance proportion of the phenotypic variation explained by the QTN. Thus, for each QTN, CV and *R*^2^ were used to describe its stability and the extent of its effect. A stable QTN is defined here as having *R*^2^ > 10% and CV < 100%.

To test QTN effect additivity, the number of QTNs with negative effect or favourable alleles (NFA) in all accessions was tallied. A simple regression of NFA on PM in the population was calculated. Correlations between the NFA and PM rating in the six datasets were calculated using the R function “*cor*”. A complete description of the post-identification QTN analysis pipeline is depicted in Appendix A.

### 4.5. Candidate Gene Prediction

A total of 1327 RGAs were identified using the RGAugury pipeline in the flax pseudomolecule [41,42]. An additional 117 disease-resistance-related genes were detected in the flax genome based on a homology search against the annotated *Arabidopsis* genes. Thus, a total of 1444 RGAs of known annotated functions to disease resistance were used for candidate gene prediction. To predict candidate resistance genes co-localized with QTNs, the RGAs located within 200 Kb of a QTN, i.e., 100 Kb on either side of the QTN, were investigated. In addition, GWAS were also performed using the models previously mentioned and the 3230 SNP subset present within 838 of the 1444 RGAs and that segregated in the germplasm under study.

### 4.6. Genomic Prediction (GP)

To gain an understanding of the best method for GP for PM resistance in flax, we evaluated the predictive ability of several marker sets: (1) all 247,160 SNPs, (2) QTNs identified by GWAS using all 247,160 SNPs, (3) QTNs within RGAs, (4) QTNs identified by GWAS using the 3230 SNPs located within RGAs, and (5) QTNs identified from all 247,160 SNPs and the 3230 SNPs located within RGAs. The following ten GP models were compared to find the optimal prediction models for PM resistance prediction: rrBLUP [76], GBLUP [77], RFR, BRR, BL, SVR [78], RKHS [76], BayesA, BayesB, and BayesC [25].

Five-fold cross-validation with 50 iterations was used to estimate the predictive ability of the models for the 372 accessions in the core collection and 75 selected breeding lines. The predictive ability (*r*) was calculated as the Pearson’s correlation coefficient between the mean genomic estimated breeding values (GEBVs) and the observed phenotypes. A custom genomic selection pipeline (GSPipeline v1.0) integrating the ten genomic prediction models implemented in the R packages rrBLUP v4.6.1 [76], BGLR v1.0.9 [79], BLR v1.6 [80], randomForest v4.7-1 [81] and sommer v4.1.6 [82] was used for GP model construction and cross-validation. Tukey’s multiple pairwise comparisons (HSD.test function) were performed to test the statistical significance of the predictive ability.

## 5. Conclusions

This study reaffirmed the quantitative nature of PM resistance in flax. The flax core collection is a valuable genetic panel that contains a broad range of genetic variation from diverse geographical origins and different improvement status, and has demonstrated, herein, that it also possesses considerable genetic variation for PM resistance. However, few accessions were highly resistant and its complementation with 75 highly resistant breeding lines was beneficial to our study. This large genetic panel with its high-density genome-wide SNPs combined with multiple single- and multi-locus GWAS models proved powerful to identify large- and minor-effect QTL, resulting in the identification of 349 QTNs and 445 candidate RGAs associated with PM resistance in flax. GWAS using the small RGA-derived SNP set further refined the candidate gene set down to 45 RGAs, which harboured QTNs within their coding sequences. Other candidate RGAs near QTNs were also supported by at least one significant SNP within their coding sequence. Significant additive features of the identified QTNs facilitate the application of these QTNs in marker-assisted and genomic selection, and a high predictive ability is expected for PM resistance. This large-scale QTL identification study provides great potential to use the identified QTNs and their potential candidate genes for PM resistance breeding and gene cloning.

## Figures and Tables

**Figure 1 ijms-23-04960-f001:**
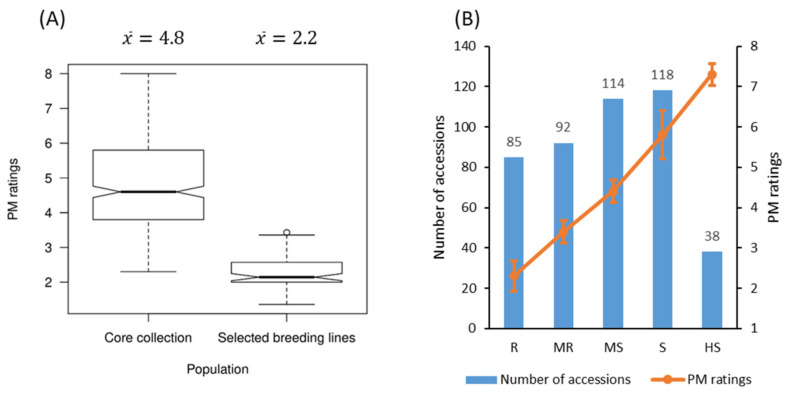
Powdery mildew (PM) field evaluation over five years (2012–2016). Box plots of PM ratings for the 372 accessions of the core collection and the 75 selected breeding lines (**A**) and the number of accessions and mean PM ratings by resistance groups (**B**). R: resistant (PM ratings of 1 to <3); MR: moderately resistant (3 to <4); MS: moderately susceptible (4 to <5); S: susceptible (5 to <7); HS: highly susceptible (7 to 9).

**Figure 2 ijms-23-04960-f002:**
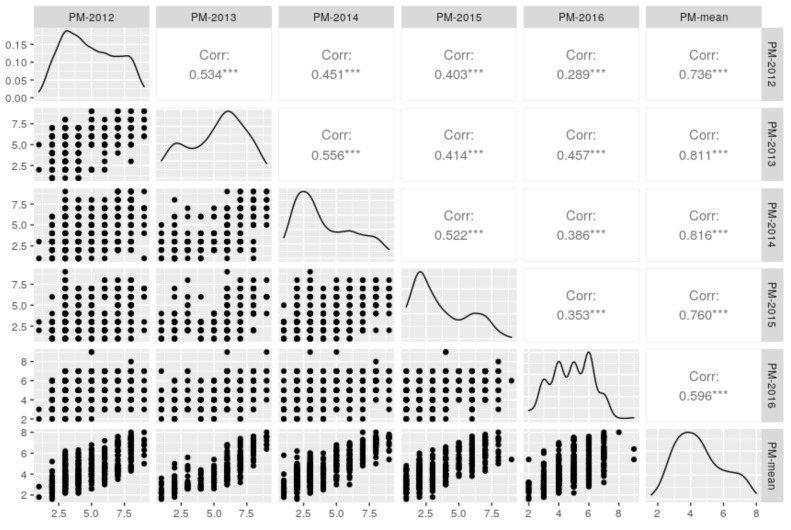
Distribution and correlation matrix of powdery mildew (PM) ratings in five consecutive years (2012–2016) and mean PM ratings over years. *** represents statistical significance at 0.001 probability level.

**Figure 3 ijms-23-04960-f003:**
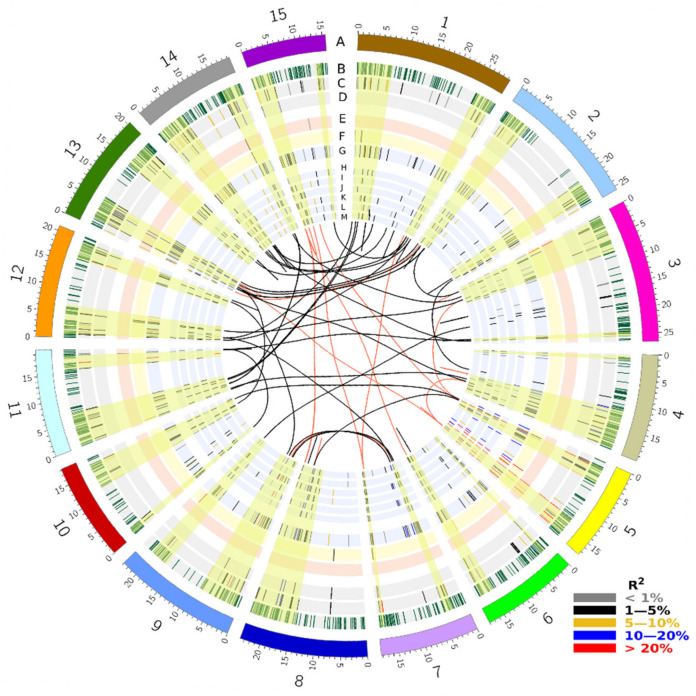
Circos map of powdery mildew (PM) resistance quantitative trait nucleotides (QTNs) and their co-located candidate genes. Track A: 15 chromosomes of the flax genome, B: 1444 resistance gene analogues (RGAs), C: 445 RGA candidates with significant SNP allele effects for PM resistance, D: 45 RGA candidates containing PM resistance QTNs, E: 9 QTNs identified by GLM, F: 15 QTNs identified by FarmCPU, G: 349 unique QTNs identified by the six multi-locus models of the mrMLM package and another multi-locus model FarmCPU, H: 68 QTNs identified by FASTmrEMMA, I: 105 QTNs identified by FASTmrMLM, J: 83 QTNs identified by ISIS_EM_BLASSO, K: 72 QTNs identified by mrMLM, L: 90 QTNs identified by pKWmEB, M: 108 QTNs identified by mrMLM_pLARmEB. The central region links the duplicated RGA candidates for the 349 QTNs identified. Allele effects of QTNs (*R*^2^) are represented by different colours as indicated in the figure legend. QTN clusters are shaded in yellow across the tracks.

**Figure 4 ijms-23-04960-f004:**
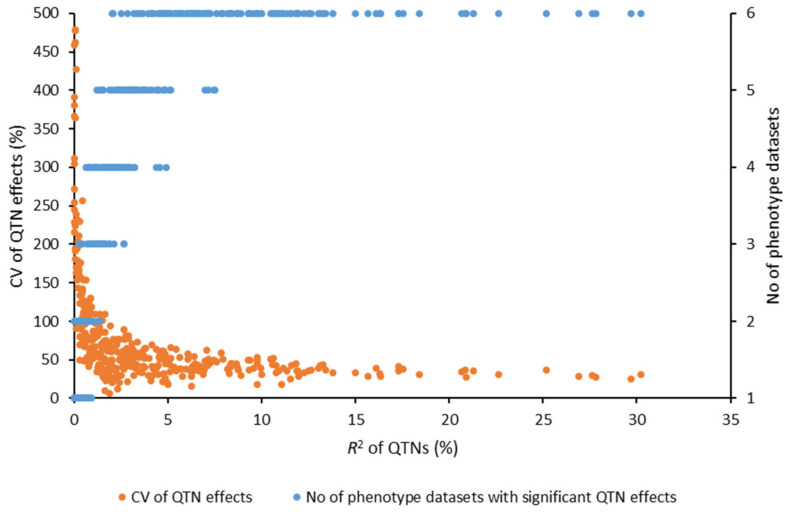
Relationship between *R*^2^, the coefficients of variation (CV, %) of quantitative trait nucleotide (QTN) effects and the number of powdery mildew (PM) datasets that showed significant QTN effects.

**Figure 5 ijms-23-04960-f005:**
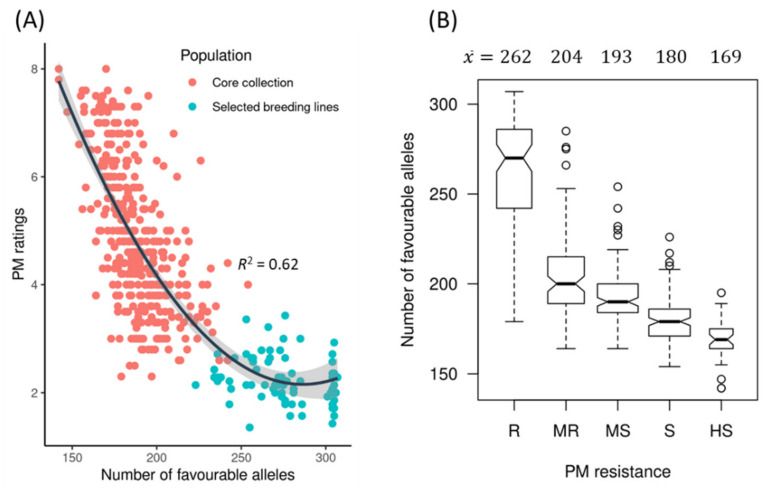
Relationships between the number of favourable alleles and powdery mildew (PM) ratings (**A**) and the PM resistance groups (**B**). The average number of favourable alleles ($\bar{x}$) per resistance group is indicated above the graph. A quadratic polynomial regression line is fitted in (**A**) with a confidence interval in grey colour.

**Figure 6 ijms-23-04960-f006:**
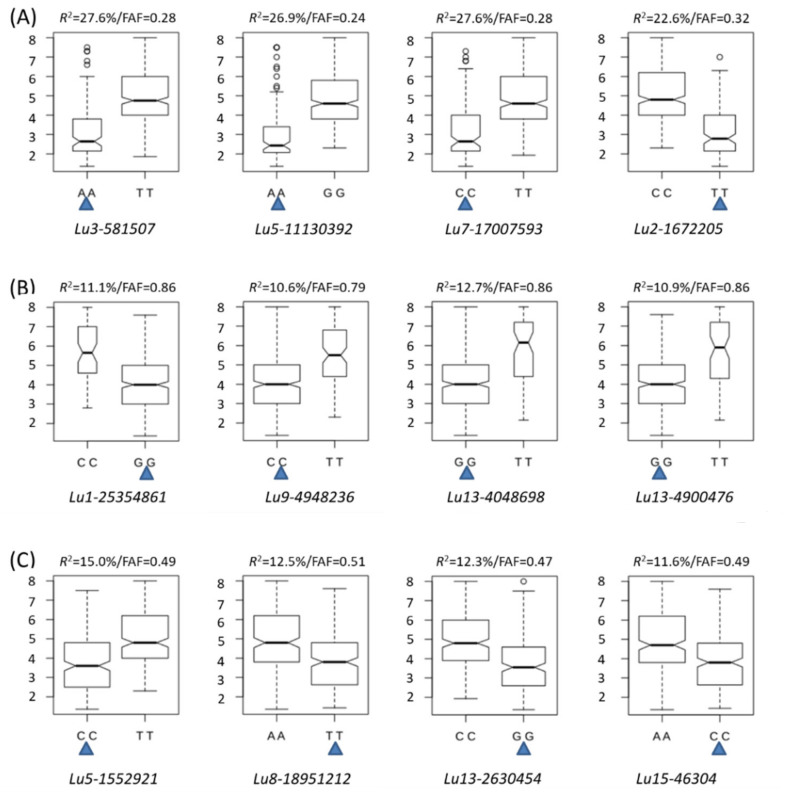
Quantitative trait nucleotide (QTN) effects of some large-effect and stable QTNs (*R*^2^ > 10%) showing QTNs with favourable allele frequencies (FAF) < 0.45 (**A**), QTNs with FAF > 0.55 (**B**) and QTNs with FAF ranging from 0.45 to 0.55 (**C**). The blue triangles indicate the favourable alleles. The QTN effects (*R*^2^) were calculated based on the PM-Mean dataset and are indicated above each box plot along with the actual FAF of the QTNs.

**Figure 7 ijms-23-04960-f007:**
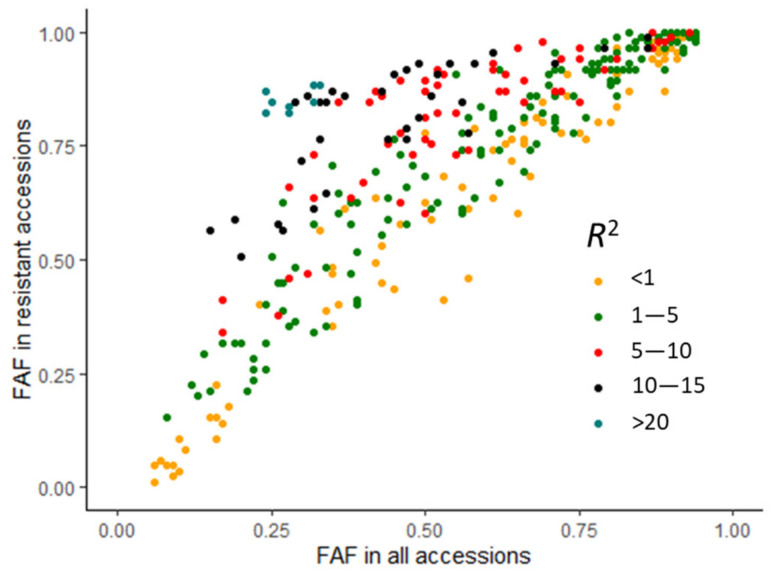
Relationship between the favourable allele frequency (FAF) in all accessions and that in the 85 resistant accession subset (R, powdery mildew ratings < 3) based on their QTN effects (*R*^2^) as colour-coded in the legend.

**Figure 8 ijms-23-04960-f008:**
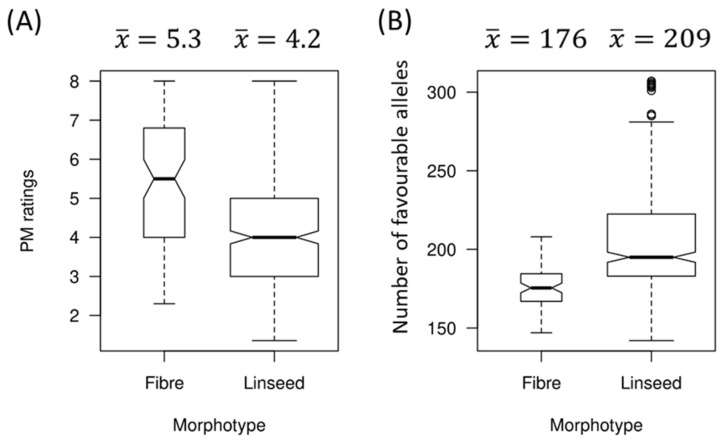
Relationships between flax morphotypes and powdery mildew (PM) ratings (**A**), and between flax morphotypes and the number of favourable alleles (**B**).

**Figure 9 ijms-23-04960-f009:**
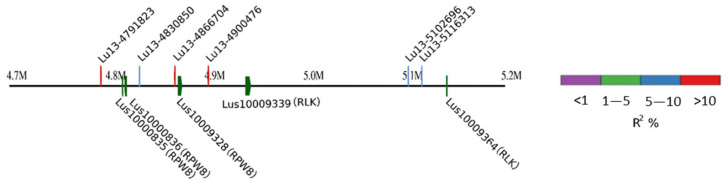
*RPW8* genes are potential candidates for four quantitative trait nucleotides (QTNs): *Lu13-4791823* (*R*^2^ = 13.79%), *Lu13-4830850* (*R*^2^ = 7.17%), *Lu13-4866704* (*R*^2^ = 12.24%), and *Lu13-4900476* (*R*^2^ = 10.9%) on chromosome 13.

**Table 1 ijms-23-04960-t001:** Powdery mildew (PM) disease ratings for 447 flax genotypes (372 accessions from a core collection and 75 selected breeding lines) from 2012 to 2016.

Year	Dataset Code	Sample Size	$\bar{x}\;\pm \;s$	Range	CV (%)
2012	PM-2012	447	4.9 ± 2.0	1.0–9.0	41.1
2013	PM-2013	253	5.2 ± 2.1	1.0–9.0	40.9
2014	PM-2014	447	4.1 ± 2.2	1.0–9.0	53.2
2015	PM-2015	391	3.6 ± 2.0	1.0–9.0	57.3
2016	PM-2016	447	4.9 ± 1.4	2.0–9.0	28.1
Mean	PM-Mean	447	4.5 ± 1.4	1.6–8.0	31.1

$\bar{x}$: population mean; *s*: standard deviation; CV: coefficient of variation. Due to missing data, only 253 and 391 accessions were available in 2013 and 2015, respectively.

**Table 2 ijms-23-04960-t002:** Quantitative trait nucleotides (QTNs) identified on 45 resistance gene analogues (RGAs) for flax powdery mildew (PM) resistance.

Gene	Chr	Gene Start Position	Gene End Position	Gene Family	Tag QTN-Position	SNP	FA	FAF	CV of QTN Effects	Effect	*R* ^2^
*Lus10006056*	1	28681727	28684082	RLK	*Lu1-28683876*	A/G	A	0.39	27.12	−0.41	4.44
*Lus10030587*	2	23955411	23958647	RLK	*Lu2-23956609*	A/C	A	0.22	38.27	−0.35	5.68
*Lus10040576*	3	5746000	5750240	TNL	*Lu3-5748445*	G/A	A	0.78	208.73	0.18	1.19
*Lus10036891*	4	12429035	12433792	WRKY	*Lu4-12432479*	G/T	G	0.37	28.39	−0.70	11.51
*Lus10041860*	4	16212942	16215322	RLK	*Lu4-16213043*	A/G	A	0.18	27.34	−0.62	3.4
*Lus10004727*	5	1534218	1535440	TNL	*Lu5-1534998*	A/G	A	0.28	31.31	−1.33	27.83
*Lus10004726*	5	1535502	1538672	TNL	*Lu5-1535619*	C/T	C	0.26	27.43	−1.54	32.6
*Lus10004719*	5	1568734	1573096	TNL	*Lu5-1569098*	G/T	G	0.31	33.14	−1.06	27.63
*Lus10032303*	5	3005188	3006765	WRKY	*Lu5-3006723*	T/C	T	0.45	22.08	−0.68	24.08
*Lus10032310*	5	3052566	3053123	DIR	*Lu5-3052714*	T/G	G	0.84	72.85	0.64	3.12
*Lus10032351*	5	3223137	3225286	RLK	*Lu5-3224350*	A/G	A	0.39	39.99	−0.64	10.62
*Lus10034795*	5	4643996	4646254	RLK	*Lu5-4646212*	C/T	C	0.49	51.12	−0.49	9.31
*Lus10029860*	5	13270430	13273781	TNL	*Lu5-13271207*	A/G	A	0.22	28.62	−1.76	34.99
*Lus10017649*	6	1878670	1885283	RLK	*Lu6-1883039*	A/G	A	0.05	36.53	−0.55	11.79
*Lus10021001*	6	15376945	15378903	RLP	*Lu6-15378264*	G/A	A	0.78	104.21	0.23	0.8
*Lus10025216*	6	16664523	16666663	WRKY	*Lu6-16666521*	C/T	T	0.58	52.26	0.12	1.13
*Lus10023199*	7	17357355	17360952	TM-CC	*Lu7-17359522*	G/A	G	0.44	61.38	−0.15	0.73
*Lus10003971*	7	17657006	17660870	TM-CC	*Lu7-17659649*	A/G	G	0.91	26.37	1.12	4.85
*Lus10021849*	8	7386368	7423024	RLK	*Lu8-7394085*	A/G	G	0.78	39.93	0.38	3.65
*Lus10022265*	8	15999079	16001545	RLK	*Lu8-15999956*	A/G	G	0.87	27.70	0.47	1.36
*Lus10007812*	8	18350700	18354799	TNL	*Lu8-18351964*	C/T	T	0.65	44.47	0.73	11.49
*Lus10002249*	8	19037487	19042009	TNL	*Lu8-19040276*	G/A	G	0.19	28.07	−1.3	15.67
*Lus10010221*	9	911825	917007	TNL	*Lu9-916748*	T/G	G	0.96	62.09	0.98	1.71
*Lus10031043*	9	6265980	6269593	RLK	*Lu9-6266682*	T/C	C	0.74	131.11	0.23	1.47
*Lus10006772*	12	719787	720446	RLK	*Lu12-720013*	A/G	G	0.80	61.75	0.5	2.61
*Lus10006732*	12	890823	894178	TNL	*Lu12-892762*	C/G	C	0.11	48.07	−0.47	1.46
*Lus10023323*	12	1893862	1897997	RLK	*Lu12-1896717*	A/T	A	0.15	25.63	−0.48	1.66
*Lus10018289*	12	5110353	5112941	TM-CC	*Lu12-5111993*	G/C	G	0.44	33.29	−0.25	2.03
*Lus10027903*	12	16614431	16619009	RLP	*Lu12-16614785*	A/G	G	0.53	53.26	0.27	20.28
*Lus10033608*	12	19122119	19129113	RLK	*Lu12-19127670*	G/A	A	0.77	93.09	0.34	2.78
*Lus10001336*	13	1747160	1750173	MLO	*Lu13-1749576*	G/A	G	0.14	41.08	−0.44	1.23
*Lus10019708*	13	4520377	4539065	TNL	*Lu13-4531367*	T/C	T	0.39	29.66	−0.73	14.08
*Lus10009364*	13	5141681	5142476	RLK	*Lu13-5142458*	C/T	C	0.18	61.62	−0.62	8.14
*Lus10030845*	13	18211794	18215062	RLK	*Lu13-18212664*	T/G	G	0.92	52.02	0.88	4.89
*Lus10028639*	14	1171345	1174215	CNL	*Lu14-1171479*	C/T	C	0.53	36.69	−0.49	10.89
*Lus10020534*	14	3457175	3462957	TNL	*Lu14-3458382*	C/G	C	0.54	23.50	−0.40	7.47
*Lus10021448*	14	4018836	4024903	TM-CC	*Lu14-4021471*	A/T	T	0.94	78.62	0.75	3.75
*Lus10014150*	14	5375403	5382647	RLK	*Lu14-5382091*	A/G	G	0.92	91.22	0.58	8.75
*Lus10015648*	14	5955689	5959658	TNL	*Lu14-5959395*	G/C	C	0.88	58.54	0.81	6.82
*Lus10015649*	14	5960004	5963280	TNL	*Lu14-5960489*	G/A	A	0.74	56.87	0.47	5.75
*Lus10008320*	14	10547787	10552007	TNL	*Lu14-10551333*	T/C	C	0.80	45.78	0.75	5.78
*Lus10035674*	14	15357234	15366290	TNL	*Lu14-15360622*	T/C	C	0.53	38.40	0.25	3.03
*Lus10039211*	14	17203248	17203932	TNL	*Lu14-17203266*	G/A	G	0.24	42.02	−1.05	14.04
*Lus10007610*	15	47907	50779	RLK	*Lu15-50397*	G/A	G	0.45	35.62	−0.52	15.63
*Lus10012678*	15	3990588	4001681	WRKY	*Lu15-3991048*	T/G	G	0.87	63.36	0.75	18.51

Chr: chromosome; FA: favourable allele; FAF: favourable allele frequency; CV: coefficient of variation; RLK: receptor-like protein kinase; RLP: receptor-like protein; TM-CC: transmembrane coiled-coil protein; NBS: nucleotide-binding site domain; LRR: leucine-rich repeat; TIR: Toll/interleukin-1 receptor-like domain; CNL: CC–NBS–LRR; TNL: TIR-NBS-LRRs; TN: TIR–NBS; TX: TIR–unknown; MLO: mildew resistance locus o.

**Table 3 ijms-23-04960-t003:** Counts of candidate resistance-related genes based on gene families and *R*^2^ of QTNs. Only candidate genes with SNPs of significant allele effects on powdery mildew (PM) ratings are included.

Candidate Gene Family	*R*^2^ of QTNs (%)					Total
	<1	1–5	5–10	10–20	>20	
DIR	1	4 (1)		1		6 (1)
DZC			1			1
EDR		1				1
MLO		5 (1)	1			6 (1)
RLK	26	65 (9)	32 (4)	13 (3)	2	138 (16)
RLP	4	8 (1)	6	2	2 (1)	22 (2)
TM-CC	4	18 (4)	3	2		27 (4)
WRKY	2	8 (1)	4	3 (2)	1 (1)	18 (4)
CNL	2	3		4 (1)	1	10 (1)
TNL	10 (1)	25 (5)	15 (2)	7 (4)	7 (4)	64 (16)
TIR		1				1
Total	49	138 (21)	62 (7)	32 (10)	13 (6)	294 (45)

The numbers in parentheses are the number of candidate genes within which QTNs were identified. RLK: receptor-like protein kinase; RLP: receptor-like protein; TM-CC: transmembrane coiled-coil protein; NBS: nucleotide-binding site domain; LRR: leucine-rich repeat; TIR: Toll/interleukin-1 receptor-like domain; CNL: CC–NBS–LRR; TNL: TIR-NBS-LRR; TN: TIR–NBS; TX: TIR–unknown; MLO: mildew resistance locus o; EDR: extreme-drug-resistant; DZC: disease resistance, zinc finger, chromosome condensation; DIR: dirigent protein.

**Table 4 ijms-23-04960-t004:** Genomic heritability (*h*^2^) and genomic predictive ability (*r*) of five datasets for powdery mildew (PM) resistance. Predictive ability was estimated based on the PM-Mean dataset and the GBLUP model.

Marker Set	*h* ^2^	*r*
349 QTNs (identified from 247,160 SNPs)	0.691 ± 0.040	0.925 ± 0.014 a
388 QTNs (identified from both 247,160 SNPs and 3230 RGA-derived SNPs)	0.698 ± 0.041	0.926 ± 0.013 a
132 QTNs located within genes (RGAs and non-RGAs)	0.481 ± 0.048	0.850 ± 0.024 b
279 QTNs/SNPs located within candidate RGAs	0.587 ± 0.050	0.822 ± 0.031 c
247,160 SNPs	0.600 ± 0.070	0.690 ± 0.049 d

Letters on the right of the predictive ability (*r*) represent statistical significance at a 5% probability level using the Tukey HSD test.

## Data Availability

All data are available in the Appendix A.

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
