# Peer review of "Insights into the Genetic Architecture and Genomic Prediction of Powdery Mildew Resistance in Flax (*Linum usitatissimum* L.)"

_ijms, 2022, doi:10.3390/ijms23094960_

Round 1

Reviewer 1 Report

The reviewed manuscript tackles with interesting topic. The article is overall well-written and -argued and in my opinion, worth publishing in its present form, once the authors address some minor aspects, which in my opinion require their attention.

  • Abbreviations should be explained on first appearance (e.g. “WRKY” in the abstract)
  • As keywords are used to assist search engines to find the article, is seems redundant to use keywords that are also apparent in the title.
  • The Y axis (vertical axis) of figure 1A, should read in the same orientation as the same axis of figure 1B. Similar issues can be found in other figures (e.g. figure 2 has to vertical text orientations, while the text sizes is fluctuating significantly among all figures).
  • Abbreviations in the text found within figures are not property explained, e.g. figure 6.

Author Response

The reviewed manuscript tackles with interesting topic. The article is overall well-written and -argued and in my opinion, worth publishing in its present form, once the authors address some minor aspects, which in my opinion require their attention.

Answer: Thank you so much for the reviewer's positive comments and support. 

  • Abbreviations should be explained on first appearance (e.g. “WRKY” in the abstract)

Answer: Thank you for pointing out this. We have expanded some abbreviations in the first appearance. Especially we did this in captions of tables and figures. By the way, it seems that WRKY is a protein domain, and it may be not an abbreviation.   

  • As keywords are used to assist search engines to find the article, is seems redundant to use keywords that are also apparent in the title.

Considering the key words might be separately used in some literature systems, we did not remove any redundancy between the title and the keywords section.    

  • The Y axis (vertical axis) of figure 1A, should read in the same orientation as the same axis of figure 1B. Similar issues can be found in other figures (e.g. figure 2 has to vertical text orientations, while the text sizes is fluctuating significantly among all figures).

Answer: Thank you so much for pointing out this. we redrew related figures: Figures 1,  5, 6, and 8. All these figures use the same text orientation for Y axis. 

  • Abbreviations in the text found within figures are not property explained, e.g. figure 6.

Answer: We reviewed all captions of the figures and tables and expanded all abbreviations. 

Reviewer 2 Report

The paper described an interesting topic of the genetic aspects for PM resistance in flax in particular to detect the genes associated to PMR. The paper is well structured and written.

Just Few point need to be checked:   

Line 22: ambiguous sentence: “… for five to eight years (2010-2017)…” please rephrase.

Line 59: Add references  wit example if possible: “..most resistance genes become ineffective after a period of time.”

Line 128: In section of Evaluation of Powdery Mildew Resistance, add justification sentence of variation of simple size between years (Table 1).

Figure 1 & 4: please improve the contrast

Please check the supplementary files: some tables (S2, S4, S5, S6, …) were mentioned on the text and don’t exist on Sup files.

Figure 4: check the axes titles “datasets"

Author Response

The paper described an interesting topic of the genetic aspects for PM resistance in flax in particular to detect the genes associated to PMR. The paper is well structured and written.

Answer: Great thanks for the reviewer's positive comments and endorsement for publication.

Just Few point need to be checked:   

Line 22: ambiguous sentence: “… for five to eight years (2010-2017)…” please rephrase.

Answer: fixed. Simply remove "five to".

Line 59: Add references  wit example if possible: “..most resistance genes become ineffective after a period of time.”

Answer: added one citation: "As a result, most resistance genes become ineffective after a period of time [8]. "

Line 128: In section of Evaluation of Powdery Mildew Resistance, add justification sentence of variation of simple size between years (Table 1).

Anmswer: added a note beneath Table 1: "Due to missing data, 153
only 253 and 391 accessions were available in 2013 and 2015, respectively"

Figure 1 & 4: please improve the contrast

Answer: Redrew Figures 1 and 4. We also redrew Figures 5, 6 and 8.

Please check the supplementary files: some tables (S2, S4, S5, S6, …) were mentioned on the text and don’t exist on Sup files.

Answer: sorry to confuse the reviewer. Thank you for pointing out this. Actually we put some supplementary tables and figures in one file, and other big tables are separate files. Now, we put each supp figure or table into a single file.

Figure 4: check the axes titles “datasets"

Answer: correct.